# Adiabatic Shortcuts Completion in Quantum Field Theory: Annihilation of Created Particles

**DOI:** 10.3390/e25091249

**Published:** 2023-08-23

**Authors:** Nicolás F. Del Grosso, Fernando C. Lombardo, Francisco D. Mazzitelli, Paula I. Villar

**Affiliations:** 1Departamento de Física, Facultad de Ciencias Exactas y Naturales, Universidad de Buenos Aires, Buenos Aires 1428, Argentina; ngrosso@df.uba.ar; 2Instituto de Física de Buenos Aires (IFIBA), CONICET—Universidad de Buenos Aires, Buenos Aires 1428, Argentina; 3Centro Atómico Bariloche and Instituto Balseiro, Comisión Nacional de Energía Atómica, Bariloche 8400, Argentina; fdmazzi@cab.cnea.gov.ar

**Keywords:** shortcuts to adiabaticity, optomechanical cavity, quantum thermodynamics

## Abstract

Shortcuts to adiabaticity (STA) are relevant in the context of quantum systems, particularly regarding their control when they are subjected to time-dependent external conditions. In this paper, we investigate the completion of a nonadiabatic evolution into a shortcut to adiabaticity for a quantum field confined within a one-dimensional cavity containing two movable mirrors. Expanding upon our prior research, we characterize the field’s state using two Moore functions that enables us to apply reverse engineering techniques in constructing the STA. Regardless of the initial evolution, we achieve a smooth extension of the Moore functions that implements the STA. This extension facilitates the computation of the mirrors’ trajectories based on the aforementioned functions. Additionally, we draw attention to the existence of a comparable problem within nonrelativistic quantum mechanics.

## 1. Introduction

Quantum thermodynamics constitutes a burgeoning research field that explores the interplay between thermodynamic principles and quantum systems [1]. By merging the foundational tenets of quantum mechanics with classical thermodynamics, it seeks to unravel the intricacies of thermal phenomena manifested at the microscopic scale. Within this insight, quantum thermal machines have emerged as pioneering devices capable of harnessing quantum effects to execute thermodynamic operations such as work extraction from heat reservoirs and refrigeration. Operating in the quantum regime, these machines exploit the distinctive attributes of quantum coherence and entanglement, thus surpassing the limitations imposed by their classical counterparts. However, to this end, it is crucial to isolate these systems from the interaction with their surroundings in order to maintain quantum correlation or even cool atoms to absolute zero.

Quantum open systems investigate the dynamics and interactions of quantum systems under the influence of an environment, accounting the reasons for which it is often challenging to isolate or completely control the quantum system [2]. These interactions introduce complexities that can lead to undesired effects, such as decoherence, dissipation, and errors. A comprehensive understanding and characterizing of the open dynamics of a system is essential for controlling it effectively and minimizing sources of errors. This knowledge allows for the design of strategies to manipulate and engineer quantum systems while mitigating the impact of unwanted interactions. In quantum thermodynamics, where precision and accuracy are of utmost importance, controlling and reducing errors is critical to achieving reliable and efficient operations.

Quantum machines play a crucial role in quantum thermodynamics by enabling the manipulation and control of quantum states and energy exchanges at the microscopic level [3,4,5]. They serve as experimental platforms for studying fundamental aspects of quantum thermodynamics. Most of the research in this area has been conducted on qubits [3] or harmonic oscillators [4] subjected to different thermodynamic cycles. In [5], a thermal machine using a quantum field subjected to an Otto cycle, implemented with a superconducting circuit (consisting of a transmission line terminated by a superconducting quantum interference device), has been considered. The performance of this machine has been studied when acting as both a heat engine and a refrigerator. It has been shown that in a nonadiabatic regime, the efficiency of the quantum cycle is affected by the dynamical Casimir effect (DCE) [6,7,8,9,10], which induces a kind of quantum friction that diminishes the efficiency. Superconducting qubits, the building blocks of circuit QED systems, provide long coherence times and high-fidelity operations and therefore offer a versatile platform for implementing thermodynamic protocols and studying quantum heat engines and refrigerators [11].

In some cases of discrete stroke quantum machines, such as a quantum harmonic oscillator or a quantum field undergoing an Otto cycle, it has been shown that the efficiency of the resulting machine is maximum for adiabatic (i.e., infinitely slow) driving [5,12]. The problem is that these conditions imply a slow evolution that can be impractical and inefficient in terms of time. Furthermore, it can lead to the loss of efficiency of a heat engine. In this scenario, a shortcut to adiabaticity (STA) appears as a promising technique to overcome the efficiency loss associated with finite-time operations and achieve results comparable to adiabatic processes. Adiabatic shortcuts is a technique that allows a system to evolve rapidly between two adiabatic states without violating the adiabatic constraints. In general, this means that for an adiabatic shortcut, no new excitations will be generated in the final state; however, it is worth noting that some STA methods, such as transitionless quantum driving [13,14], also ensure that nonadiabatic excitations are suppressed even at intermediate times. Other methods for implementing shortcuts to adiabaticity include the use of invariants [15], fast forward techniques [16], optimal protocols [17], fast quasiadiabatic (FAQUAD), etc. [18].

STA has been considered from a theoretical and/or an experimental point of view for different physical systems: trapped ions [19], cold atoms [20], ultracold Fermi gases [21], Bose–Einstein condensates in atom chips [22], spin systems [23], etc. STA has been also proposed to relieve the trade-off of efficiency and power [24,25,26], both in single-particle quantum heat engines (QHEs) [27] and in many-particle QHEs [28,29,30].

In a previous work [31], we explored the possibility of applying STA in quantum field theory. Particularly, we showed how to implement an STA for a massless scalar field inside a cavity with a moving wall in (1+1) dimensions. The approach is based on the already known solution to the problem by exploiting the conformal symmetry. The shortcuts take place whenever the solution matches the adiabatic Wentzel–Kramers–Brillouin (WKB) solution [32] and there is no DCE. In [33], we generalized the results of a quantum scalar field in a one-dimensional optomechanical cavity to two moving mirrors. We showed that given the trajectories for the left and right mirrors, it is possible to find an STA ruled by the effective trajectories of the mirrors. When implemented in finite time, these trajectories result in the same state as if the original ones had been evolved adiabatically. This protocol has the advantage that it can be easily implemented experimentally using either an optomechanical cavity or superconducting circuits, as it does not require additional exotic potentials. Moreover, the effective trajectory can be computed from the original one quite simply, paving the way for more efficient quantum field thermal machines.

In this context, herein, we find a general approach to complete an STA in the optomechanical cavity. By completing an STA, we refer to the following scenario: let us consider the system initially in its ground state; then, subject it to a time-dependent, nonadiabatic transformation. As a result of this transformation, the system transitions to an excited state. The arising question is if it is feasible to carry out a subsequent transformation in a manner that leads the system back to its ground state. If such a possibility exists, we refer to it as an STA completion. This completion provides an additional tool for the control of quantum systems.

The paper is organized as follows. In Section 2, we review the main results for the quantization of a massless scalar field in a cavity with two moving boundaries. The excitation of the system can be described in terms of the so-called Moore’s functions [6,34], which are the main tools to construct the STA for the field. Before discussing the STA completion for this system, and as a warm-up, in Section 3, we describe a simple analogy using a quantum harmonic oscillator with time-dependent frequency. We see that there is a simple way to unfold the evolution and construct an STA by an inverse engineering method based on a smooth continuation of the so-called Ermakov function [35]. The striking similarity between the Ermakov and Moore functions is used in Section 4 to construct the STA completion in the optomechanical cavity. We show that there is a general procedure to build up STA completions and that in some particular cases, the protocol is extremely simple and shows time-inversion invariance. Section 5 contains the conclusions of our work.

## 2. The Optomechanical Cavity

The system we consider is a scalar field, Φ(x,t), inside a one-dimensional cavity delimited by a moving mirror at each end whose positions are given by L(t) and R(t), respectively (see Figure 1). The evolution of the field is determined by the wave equation inside the cavity
(1)$$\left(\partial_{x}^{2}-\partial_{t}^{2}\right)\Phi \left(x,t\right)=0,$$
and Dirichlet boundary conditions on each mirror
(2)Φ(L(t),t)=Φ(R(t),t)=0.

Here, and in the rest of the paper, we use units where $c=\hbar =k_{B}=1$.

It is known that the time evolution of the field is solved by expanding the field in modes
(3)$$\Phi \left(x,t\right)=\sum_{k=1}^{\infty }\left[a_{k}\psi_{k}\left(x,t\right)+a_{k}^{†}\psi_{k}^{*}\left(x,t\right)\right],$$
where the modes are given by [34]
(4)$$\psi_{k}\left(x,t\right)=\frac{i}{\sqrt{4\pi k}}\left[e^{-ik\pi G(t+x)}+e^{ik\pi F(t-x)}\right].$$Here, F(z) and G(z) are functions determined by the so-called Moore’s equations
(5)$$\begin{array}{cc} G(t+L(t))-F(t-L(t)) & =0 \end{array}$$
(6)$$\begin{array}{cc} G(t+R(t))-F(t-R(t)) & =2. \end{array}$$The functions F(z) and G(z) implement the conformal transformation
(7)$$\bar{t}+\bar{x}=G\left(t+x\right)\;\bar{t}-\bar{x}=F\left(t-x\right)$$
such that, in the new coordinates, the left and right mirrors are static at ${\bar{x}}_{L}=0$ and ${\bar{x}}_{R}=1$. In the particular case in which the left mirror is static at x=0, we have G(t)=F(t) and a single nontrivial Moore equation.

The description of the dynamics of the quantum field in the presence of moving mirrors is therefore reduced to solving the Moore’s equations. Once this is achieved, the renormalized energy density of the field can be obtained from [34]
(8)$$\begin{array}{c} {\langle T_{tt}\left(x,t\right)\rangle }_{\mathrm{ren}}=f_{G}\left(t+x\right)+f_{F}\left(t-x\right), \end{array}$$
where
(9)$$\begin{array}{ccc} f_{G} & = & -\frac{1}{24\pi }\left[\frac{G^{\prime \prime \prime }}{G^{\prime }}-\frac{3}{2}{\left(\frac{G^{\prime \prime }}{G^{\prime }}\right)}^{2}\right]+\frac{{\left(G^{\prime }\right)}^{2}}{2}\left[-\frac{\pi }{24}+Z\left(Td_{0}\right)\right]\\ f_{F} & = & -\frac{1}{24\pi }\left[\frac{F^{\prime \prime \prime }}{F^{\prime }}-\frac{3}{2}{\left(\frac{F^{\prime \prime }}{F^{\prime }}\right)}^{2}\right]+\frac{{\left(F^{\prime }\right)}^{2}}{2}\left[-\frac{\pi }{24}+Z\left(Td_{0}\right)\right], \end{array}$$
and $d_{0}=\left|R\left(0\right)-L\left(0\right)\right|$ is the initial length of the cavity. The above result is valid when the state of the field is initially in a thermal state at temperature *T* and $Z(Td_{0})$ is related to the initial mean energy
(10)$$Z\left(Td_{0}\right)=\sum_{n=1}^{\infty }\frac{n\pi }{\exp\left(\frac{n\pi }{Td_{0}}\right)-1}.$$The expression for the renormalized energy–momentum tensor above can be obtained using the standard approach based on point-splitting regularization (see for instance [36]). It can also be derived using the conformal anomaly associated with the conformal transformation Equation (Equation 7) [37]. In the rest of the paper, we consider the T=0 case, in which the field is initially in the vacuum state.

It is important to note that for a static cavity with L(t)=0, $R\left(t\right)=d_{0}$, the general solution for F(z) and G(z) is
(11)$$\begin{array}{c} F\left(z\right)=G\left(z\right)=\frac{\left(z-z_{0}\right)}{d_{0}}+p\left(t\right), \end{array}$$
where $z_{0}$ is a constant, and p(t) is a $2d_{0}$-periodic function. However, a closer look at Equation (Equation 8) shows that the renormalized energy density reduces to the vacuum energy (the static Casimir energy density) if and only if p(t)=0, in other words, if the Moore functions are linear. Thus, it is the initial state of the cavity that determines this function, and the phenomenon of particle creation appears when F(z) and G(z) are nonlinear functions. The periodic function contains all the information of the excited state of the field.

### 2.1. STA for the Field

It is particularly challenging to find an STA for the quantum field in the cavity, since the only parameters that we can control and that affect the time evolution of the field are the positions of the left and right walls, L(t) and R(t), respectively. However, in previous papers [31,33], we have shown that this is indeed possible and can be achieved as follows. First, we find the adiabatic Moore functions
(12)$$\begin{array}{cc} F_{\mathrm{ad}}\left(t\right) & =\int dt\frac{1}{R_{\mathrm{ref}}\left(t\right)-L_{\mathrm{ref}}\left(t\right)}+\frac{1}{2}\frac{R_{\mathrm{ref}}\left(t\right)+L_{\mathrm{ref}}\left(t\right)}{R_{\mathrm{ref}}\left(t\right)-L_{\mathrm{ref}}\left(t\right)} \end{array}$$
(13)$$\begin{array}{cc} G_{\mathrm{ad}}\left(t\right) & =\int dt\frac{1}{R_{\mathrm{ref}}\left(t\right)-L_{\mathrm{ref}}\left(t\right)}-\frac{1}{2}\frac{R_{\mathrm{ref}}\left(t\right)+L_{\mathrm{ref}}\left(t\right)}{R_{\mathrm{ref}}\left(t\right)-L_{\mathrm{ref}}\left(t\right)}. \end{array}$$
which correspond to the infinitely slow evolution of the field for reference trajectories $L_{\mathrm{ref}}\left(t\right)$ and $R_{\mathrm{ref}}\left(t\right)$. Then, we look for effective trajectories $L_{\mathrm{eff}}\left(t\right)$ and $R_{\mathrm{eff}}\left(t\right)$ such that they give rise to the adiabatic Moore functions previously found
(14)$$\begin{array}{cc} G_{\mathrm{ad}}\left(t+L_{\mathrm{eff}}\left(t\right)\right)-F_{\mathrm{ad}}\left(t-L_{\mathrm{eff}}\left(t\right)\right) & =0 \end{array}$$
(15)$$\begin{array}{cc} G_{\mathrm{ad}}\left(t+R_{\mathrm{eff}}\left(t\right)\right)-F_{\mathrm{ad}}\left(t-R_{\mathrm{eff}}\left(t\right)\right) & =2. \end{array}$$The effective trajectories obtained produce an evolution of the field in finite time that at the end replicates the adiabatic one for the reference trajectories; hence, they constitute an STA.

This protocol has the potential to dramatically improve the efficiency of a thermodynamical cycle, but the energy cost of the shortcut should be taken into account [24,25,26,32]. Although there is no universal consensus on exactly how to measure this cost, one possible metric in standard quantum mechanics is given by
(16)$$\langle \delta W\rangle =\frac{1}{\tau }\int_{0}^{\tau }\left[\langle H_{\mathrm{eff}}\left(t\right)\rangle -\langle H_{\mathrm{ref}}\left(t\right)\rangle \right]dt,$$
where $H_{\mathrm{eff}}$ is the Hamiltonian that implements a shortcut to the adiabatic evolution for a reference Hamiltonian $H_{\mathrm{ref}}$, and protocols have a duration τ. However, in quantum field theory, the reference and effective protocols have different durations, $\tau_{\mathrm{ref}}$ and $\tau_{\mathrm{eff}}$, respectively, and so the previous measure should be adapted. The simplest possible generalization of the energy cost to QFT would be
(17)$$\langle \delta W\rangle =\frac{1}{\tau_{\mathrm{eff}}}\int_{0}^{\tau_{\mathrm{eff}}}\int_{L_{\mathrm{eff}}\left(t\right)}^{R_{\mathrm{eff}}\left(t\right)}{\langle T_{tt}^{\mathrm{eff}}\rangle }_{\mathrm{ren}}dxdt-\frac{1}{\tau_{\mathrm{ref}}}\int_{0}^{\tau_{\mathrm{ref}}}\int_{L_{\mathrm{ref}}\left(t\right)}^{R_{\mathrm{ref}}\left(t\right)}{\langle T_{tt}^{\mathrm{ref}}\rangle }_{\mathrm{ren}}dxdt,$$
but further investigations are needed in order to understand whether this is in fact a faithful measure.

We discuss STA completions for this system in Section 4. Before doing that, we analyze the same problem in a simpler context that serves to illustrate the reverse engineering method used to build the STA completions.

## 3. A Simple Analogue: The Ermakov Equation for the Harmonic Oscillator with Time-Dependent Frequency

In this Section, we describe a simple analogue of the STA in QFT using a quantum harmonic oscillator. As we see, the excitation of the harmonic oscillator caused by the time dependence of the frequency can be described by the so-called Ermakov function. This function exhibits a behavior similar to that of the Moore functions for the scalar field in the optomechanical cavity.

The dynamics of a harmonic oscillator with a time-dependent frequency are given by
(18)$$\ddot{q}+\omega^{2}\left(t\right)q=0\;.$$The position operator $\hat{q}$ can be written in terms of annihilation and creation operators ($\hat{a}$ and ${\hat{a}}^{†}$) as
(19)$$\hat{q}\left(t\right)=q\left(t\right)\hat{a}+q^{*}\left(t\right){\hat{a}}^{†}\;,$$
where q(t) is a solution of Equation (Equation 18) with Wronskian given by
(20)$$\dot{q}q^{*}-{\dot{q}}^{*}q=i\;.$$The Wronskian condition implies that the solutions can be written in terms of a real function W(t) as
(21)$$q\left(t\right)=\frac{1}{\sqrt{2W(t)}}e^{i\int^{t}W\left(t^{\prime }\right)dt^{\prime }}$$
that satisfies the equation
(22)$$\omega^{2}=W^{2}+\frac{1}{2}\left(\frac{\ddot{W}}{W}-\frac{3}{2}{\left(\frac{\dot{W}}{W}\right)}^{2}\right)\;,$$
which is equivalent to Equation (Equation 18).

For a slowly varying function ω(t), we have W≃ω, and Equation (Equation 21) gives the usual lowest-order WKB solution. Equation (Equation 22) can be used to obtain the higher-order corrections by solving it recursively using an expansion in the number of derivatives of ω. Alternatively, one can use an inverse engineering approach and think of Equation (Equation 21) as the exact solution of the problem with a frequency $\omega^{2}$ given by Equation (Equation 22).

Assuming that the frequency tends to constants values $\omega^{\mathrm{in},\mathrm{out}}$ for t→±∞, and that the oscillator is in the ground state $|0_{\mathrm{in}}\rangle$ for t→−∞, in the case of a nonadiabatic evolution the oscillator will be excited for t→+∞, that is $|\langle 0_{\mathrm{out}}|0_{\mathrm{in}}\rangle |\neq 1$. The in and out basis are the solutions of Equation (Equation 21) that satisfy
(23)$$q^{\mathrm{in},\mathrm{out}}\left(t\right)\underset{t\to \pm \infty }{\to }\frac{1}{\sqrt{2\omega^{\mathrm{in},\mathrm{out}}}}e^{-i\omega^{\mathrm{in},\mathrm{out}}t}\;.$$

The Bogoliubov transformation that connects the in and out basis and in and out Fock spaces, when nontrivial, is an indication of the excitation of the system due to the external time dependence:(24)$$\begin{array}{ccc} q^{\mathrm{out}} & = & \alpha \;q^{\mathrm{in}}+\beta \;q^{\mathrm{in}\;*}\\ {\hat{a}}^{\mathrm{out}} & = & \alpha^{*}\;{\hat{a}}^{\mathrm{in}}-\beta^{*}\;{\hat{a}}^{\mathrm{in}\;†}\;. \end{array}$$As is well known, and described in more detail below, the in vacuum can be written as a squeezed state in terms of the out states.

A frequency ω(t) that leads to an evolution that does not produce an excitation of the harmonic oscillator constitutes an STA. It is important to remark that the evolution at intermediate times is in general nonadiabatic, but the system returns to the initial state when the effective frequency becomes constant at t→+∞. The system is excited at intermediate times and subsequently returns to its ground state.

### 3.1. The Lewis-Riesenfeld Approach and the Ermakov Equation

In the Lewis–Riesenfeld approach, the solutions to Equation (Equation 18) are written in terms of the so-called Ermakov function $\rho =1/\sqrt{W}$ that satisfies the Ermakov equation
(25)$$\ddot{\rho }+\omega^{2}\left(t\right)\rho -\frac{1}{\rho^{3}}=0\;,$$
which is equivalent to Equation (Equation 22).

It is possible to show that within a temporal interval where $\omega =\omega_{0}$ is constant, the general solution of the Ermakov equation reads [38,39]
(26)$$\rho^{2}\left(t\right)=\frac{1}{\omega_{0}}\left[\cosh\delta -\sinh\delta \sin(2\omega_{0}t+\varphi )\right]\;,$$
where δ and φ are arbitrary constants. For δ=0, we have the usual solution for the harmonic oscillator with frequency $\omega_{0}$. If the frequency is $\omega_{0}$ for t<0, time-dependent in the interval 0<t<τ, and then stops at $\omega_{1}$, for t<0 we will have $\rho^{2}=1/\omega_{0}$, and at the end of the motion, for t>τ, $\rho^{2}$ is given by Equation (Equation 26) with $\omega_{0}\to \omega_{1}$. The values of δ and φ will depend on the whole temporal evolution of the frequency. In general, the oscillator will end up in a squeezed state. In ref. [40], it has been shown that for the particular case $\omega_{0}=\omega_{1}$, different evolutions ω(t) may lead to the same Ermakov function for t>τ and therefore to the same excited state. If the evolution is such that ρ is also constant for t>τ, then we have an unexciting evolution.

One can use reverse engineering to find effective unexciting evolutions $\omega_{\mathrm{eff}}^{2}\left(t\right)$ by an adequate choice of $\rho_{\mathrm{ref}}^{2}=1/\omega_{\mathrm{ref}}\left(t\right)$; indeed, assuming that $\rho_{\mathrm{ref}}^{2}$ is constant both for t<0 and t>τ, and plugging this "reference" Ermakov function into Equation (Equation 25), one can obtain the effective evolution as
(27)$$\omega_{\mathrm{eff}}^{2}=\frac{1}{\rho_{\mathrm{ref}}^{4}}-\frac{{\ddot{\rho }}_{\mathrm{ref}}}{\rho_{\mathrm{ref}}}\;.$$Note that for this effective evolution the function q(t) evolves as the adiabatic solution for the reference frequency $\omega_{\mathrm{ref}}$ in a finite time. The system may admit, or not, situations where $\omega_{\mathrm{eff}}^{2}\left(t\right)<0$; so, one should choose $\rho_{\mathrm{ref}}\left(t\right)$ appropriately in models where this is physically unacceptable.

### 3.2. De-Excitation of the Harmonic Oscillator

Now, we address the following question: assume that the frequency is equal to $\omega_{-}$ for t<0 and evolves from $\omega_{-}$ to $\omega_{+}$ during the interval $0<t<\tau_{1}$ in such a way that the evolution “generates particles”, that is, that the oscillator is in an excited state for $t>\tau_{1}$. We denote by $\omega_{I}\left(t\right)$ the function that interpolates between $\omega_{-}$ and $\omega_{+}$. Is it possible to find a subsequent time evolution of the frequency, $\omega_{\mathrm{II}}\left(t\right)$, in an interval $t_{1}<t<\tau_{2}$ such that the final frequency is $\omega_{++}$ and that the final state of the oscillator is $|0_{++}\rangle$? In other words, we are looking for a complementary time-dependent function $\omega_{\mathrm{II}}\left(t\right)$ such that the joint effective protocol
(28)$$\omega_{\mathrm{eff}}\left(t\right)=\left\{\begin{array}{cc} \begin{array}{c} \omega_{-}\\ \omega_{I}\left(t\right)\\ \omega_{+}\\ \omega_{\mathrm{II}}\left(t\right)\\ \omega_{++} \end{array} & \begin{array}{c} t<0\\ 0<t<\tau_{1}\\ \tau_{1}<t<t_{1}\\ t_{1}<t<t_{1}+\tau_{2}\\ t_{1}+\tau_{2}<t \end{array} \end{array}\right.,$$
converts an initially nonadiabatic evolution into an STA.

After the initial evolution (described by $\omega_{I}\left(t\right)$), one can prove that the initial vacuum state becomes a squeezed state. That is, for $\tau_{1}<t<t_{1}$, we have [41]
(29)$$|0_{-}\rangle =c_{0}\sum_{n\geq 0}{\left(-\frac{\beta^{*}}{\alpha }\right)}^{n}\frac{\sqrt{(2n)!}}{2^{n}n!}|2\;n_{+}\rangle \;,$$
where α and β are the coefficients of the Bogoliubov transformation. The mean occupation number of the + states reads
(30)$$\langle 0_{-}|a_{+}^{†}\;a_{+}|0_{-}{\rangle =|\beta |}^{2}\;.$$

In order to unfold the evolution and generate the corresponding antisqueezing, one could choose an adequate Ermakov function as follows: for $t<\tau_{1}\;,\;\rho \left(t\right)$ is determined by $\omega_{I}\left(t\right)$. It is an oscillating function for $\tau_{1}<t<t_{1}$. We can now consider a smooth continuation of this function that starts at $t_{1}$ and becomes constant $\rho^{2}\left(t\right)=1/\omega_{++}$ for $t>t_{1}+\tau_{2}$. From the complete Ermakov function, one can determine the evolution of the “de-exciting frequency” $\omega_{\mathrm{II}}\left(t\right)$ in the interval $t_{1}<t<t_{1}+\tau_{2}$ that interpolates between $\omega_{+}$ and $\omega_{++}$. The combination of the two evolutions implements the STA.

When $\omega_{-}=\omega_{++}$, the second evolution $\omega_{\mathrm{II}}\left(t\right)$ can be chosen to be the time reversal of the first evolution $\omega_{I}\left(t\right)$. This symmetric trajectory always exists and can be constructed as follows. After the first evolution, for $t>\tau_{1}$, the square of the Ermakov function is given by Equation (Equation 26). At any time $t_{n}$ that corresponds to a maximum or minimum of this periodic function, one can extend the Ermakov function symmetrically, that is $\rho_{\mathrm{II}}\left(t\right)=\rho_{I}\left(2t_{n}-t\right)$ for $t>t_{n}$. From the Ermakov Equation (Equation 25), one can easily show that the whole evolution of the frequency is time-symmetric around $t_{n}$.

It is worth remarking that the temporal reverse of an adiabatic shortcut is also an adiabatic shortcut, as can be easily checked using the Ermakov equation. We show a similar property for the moving mirrors in the next section.

Finally, we point out that for a general state, the unfolding would not be possible, i.e., given an arbitrary state with the same ${\left|\beta \right|}^{2}$, an unitary evolution will not lead to a vacuum state. Note that a given value of ${\left|\beta \right|}^{2}$ only gives the mean occupation number but does not have the information about the full quantum state of the oscillator.

## 4. Completing an STA in the Optomechanical Cavity

In this section, we come back to the STA in the optomechanical cavity. We present different alternatives for completing a given trajectory into an STA for the quantum field. That is, we assume the cavity was initially in a vacuum state (zero temperature) at position $R_{-}$ and has suffered a perturbation that moved the right wall according to the trajectory $R_{I}\left(t\right)$ with an associated Moore function F(z). Our goal will be to find a second trajectory $R_{\mathrm{II}}\left(t\right)$ such that the joint trajectory $R_{\mathrm{eff}}\left(t\right)$ has a Moore function that is linear at early and late times, which will result in an adiabatic evolution of the field. We present different strategies to achieve this based on the same ideas described in the previous section for the quantum harmonic oscillator.

Before proceeding, we need to establish a magnitude to decide whether an STA has been achieved and measure how far we are from one. Hence, we define the adiabaticity coefficient
(31)$$Q\left(t\right):=\frac{E(t)}{E_{\mathrm{ad}}\left(t\right)},$$
where E(t) is the total energy in the cavity
(32)$$\begin{array}{c} E\left(t\right)=\int_{L(t)}^{R(t)}dx{\langle T_{00}\left(x,t\right)\rangle }_{\mathrm{ren}}, \end{array}$$
while the adiabatic energy is given by
(33)$$\begin{array}{c} E_{\mathrm{ad}}\left(t\right)=-\frac{\pi }{24d}, \end{array}$$
where d=|R(t)−L(t)| is the length of the cavity. We are assuming that the field is initially in the vacuum state.

Once the effective trajectories that complete a shortcut and the associated Moore functions are obtained, the energy and adiabaticity coefficients can then be calculated using Equations (Equation 8) and (Equation 32).

### 4.1. Reverse Engineering

We now present a first method for completing a trajectory to be an adiabatic shortcut. The method is theoretically quite simple and works for an arbitrary final position. Its practical implementation is not so simple, and it involves three steps: the computation of the Moore functions associated with the initial evolution, the extension of them smoothly into linear functions, and the computation of the effective trajectory using inverse engineering.

As mentioned, our goal here is to complete an adiabatic shortcut for a cavity that was initially in a vacuum state and was perturbed by an arbitrary trajectory of one of the mirrors that left it in an excited state. Because of the initial condition, we know that the associated Moore function was a linear function before the motion started and, since an STA is achieved by having a Moore function that is linear at early and late times, we can theoretically complete an adiabatic shortcut by simply extending the Moore function continuously to a linear function at late times and computing the associated effective trajectory. Notice that this method for completing an STA is completely analogous to the one previously presented for the harmonic oscillator.

In order to illustrate this strategy, let us consider an initial trajectory for the mirrors given by
(34)$$\begin{array}{cc} L_{I}\left(t\right) & =0\\ R_{I}\left(t\right) & =\left\{\begin{array}{cc} \begin{array}{c} R_{-}\\ f(t)\\ R_{+} \end{array} & \begin{array}{c} t<0\\ 0<t<\tau \\ \tau <t \end{array} \end{array}\right., \end{array}$$
where f(t) is given by the following polynomial
(35)$$\begin{array}{cc} f(t) & =R_{-}\left(1-\epsilon \delta \left(t/\tau \right)\right)\\ \delta (x) & =35x^{4}-84x^{5}+70x^{6}-20x^{7}, \end{array}$$
which verifies δ(0)=0 and δ(1)=1. The choice of the polynomial that defines δ(t) ensures that $R_{I}\left(t\right)$ and its first three derivatives are continuous. This is needed to avoid spurious divergences in the energy–momentum tensor (see Equation (Equation 9)).

Using this trajectory, we can compute the associated Moore function $F_{I}$, which is oscillating at late times due to the creation of photons, and extend it continuously into a linear function $F_{\mathrm{eff}}$ with any desired slope (which in turn determines the final position of the boundary). Then, we can recover the corresponding trajectory of the mirror $R_{\mathrm{eff}}\left(t\right)$ for the extended function by solving Equation (6) (Figure 2).

The resulting trajectory is a well-defined continuous function that starts at the final position of the initial perturbation and ends at the position set by us through the slope of the linear function at late times. The speed can be seen to be below the speed of light. We can also check that the end-to-end trajectory constitutes a shortcut, since the adiabaticity parameter $Q_{\mathrm{eff}}$ starts and ends at 1 (Figure 3).

The disadvantage of this method is that in order to erase the photons generated by the initial motion, one needs to compute the Moore function of the field, then extend it smoothly, and subsequently compute the trajectory that completes an adiabatic shortcut by solving Moore’s equation.

### 4.2. Short Pulses

In this section, we explore how to complete an STA using the idea of the previous section, now applied to an arbitrary short pulse. In other words, we show how to erase the photons generated by any brief motion of one of the cavity mirrors and reset the cavity to its initial state.

We again consider an initial trajectory given by Equations (Equation 34) and (Equation 35) such that $\tau \leq R_{\pm }$. It can be seen that in this case, the derivative of the Moore function, $F_{I}^{\prime }\left(z\right)$, has a simple structure. This is given by an initial constant, $1/R_{-}$, followed by a pulse that starts at $z=R_{-}$ and ends at $z=\tau +R_{+}$. From there, and up to $z=2R_{+}$, the function again takes the value of the initial constant. This structure is repeated periodically with period $2R_{+}$ (Figure 4).

In order to show this, we consider the derivative of the Moore Equation (6)
(36)$$F_{I}^{\prime }\left[t+R\left(t\right)\right]\left[1+{\dot{R}}_{I}\left(t\right)\right]-F_{I}^{\prime }\left[t-R_{I}\left(t\right)\right]\left[1-{\dot{R}}_{I}\left(t\right)\right]=0,\;\forall t.$$Using the initial condition $R_{I}\left(t<0\right)=R_{-}$ and $F_{I}^{\prime }\left(z<0\right)=1/R_{-}$, we have
(37)$$F_{I}^{\prime }\left[t+R_{I}\left(t\right)\right]=\frac{1}{R_{-}}\frac{\left[1-{\dot{R}}_{I}\left(t\right)\right]}{\left[1+{\dot{R}}_{I}\left(t\right)\right]}=\frac{1}{R_{-}}\;\mathrm{if}\;t<0$$
from which we conclude that the derivative of the Moore function is constant up to $z<R_{-}$
(38)$$F_{I}^{\prime }\left[z\right]=\frac{1}{R_{-}}\;\mathrm{if}\;z<R_{-}.$$Additionally, we can use Equation (Equation 37) to show that if $0<t<\tau <R_{\pm }$, then t−R(t)<0, and we have
(39)$$F_{I}^{\prime }\left[t+R_{I}\left(t\right)\right]=\frac{1}{R_{-}}\frac{\left[1-{\dot{R}}_{I}\left(t\right)\right]}{\left[1+{\dot{R}}_{I}\left(t\right)\right]}\;\mathrm{if}\;0<t<\tau .$$This equation sets the shape of $F_{I}^{\prime }$ for $R_{-}<z=t+R_{I}\left(t\right)<\tau +R_{+}$. Lastly, once the trajectory has stopped, we have
(40)$$F_{I}^{\prime }\left[t+R_{+}\right]=F_{I}^{\prime }\left[t-R_{+}\right]\;\mathrm{if}\;\tau <t,$$
which relates the $F_{I}^{\prime }$ at late times to its value at an earlier time. Indeed, we can rewrite
(41)$$F_{I}^{\prime }\left[z\right]=F_{I}^{\prime }\left[z-2R_{+}\right]\;\mathrm{if}\;\tau +R_{+}<t+R_{+}=z,$$
and this sets the $2R_{+}$ periodicity at late times. We can use this to find the value of the derivative for $z<R_{-}+2R_{+}$ using Equation (Equation 38) to find
(42)$$F_{I}^{\prime }\left[z\right]=\frac{1}{R_{-}}\;\mathrm{if}\;\tau +R_{+}<z<R_{-}+2R_{+},$$
which establishes that the derivative of the Moore function is constant in that interval.

From this structure, it is very easy to find an extension of the Moore function that is linear at late times, taking advantage of the fact that the derivative of the Moore function is constant in the interval $\tau +R_{+}<z<R_{-}+2R_{+}$. The natural extension is to complete the derivative of the Moore function, assuming that is constant for $t>R_{-}+2R_{+}$. Therefore, $F_{\mathrm{eff}}\left(z\right)=z/R_{-}$ for $z>R_{-}+2R_{+}$. The extended Moore function will give rise to a trajectory $R_{\mathrm{eff}}\left(t\right)$ that will coincide with $R_{I}\left(t\right)$ initially; then, the trajectory will be constant, and finally, there will be an erasing trajectory $R_{\mathrm{II}}\left(t\right)$ which will have to come back to $R_{-}$ to satisfy the final slope of the Moore function.

This erasing trajectory will satisfy Equation (Equation 36)
(43)$$\frac{1}{R_{-}}\left(\frac{1+{\dot{R}}_{\mathrm{II}}\left(t\right)}{1-{\dot{R}}_{\mathrm{II}}\left(t\right)}\right)=F_{\mathrm{eff}}^{\prime }\left[t-R_{\mathrm{II}}\left(t\right)\right],\;\mathrm{if}\;t>R_{-}+R_{+}$$
where we used the extension condition $F_{\mathrm{eff}}^{\prime }\left(t+R_{\mathrm{eff}}\left(t\right)\right)=1/R_{-}$ for $z=t+R_{\mathrm{eff}}\left(t\right)>R_{-}+2R_{+}$. Of course, this equation should be coupled with the condition that the erasing trajectory begins where $R_{I}$ ended, i.e.,
(44)$$R_{\mathrm{II}}\left(t=R_{+}+R_{-}\right)=R_{+}.$$The previous equation can be solved exactly in terms of the initial trajectory by taking
(45)$$R_{\mathrm{II}}\left(t\right)=R_{+}-\left[R_{I}\left(t-t_{1}\right)-R_{-}\right],\mathrm{if}\;t_{1}<t<t_{1}+\tau$$
where $t_{1}=R_{-}+R_{+}$.

This can be seen by replacing in Equation (Equation 43)
(46)$$\frac{1}{R_{-}}\left(\frac{1-{\dot{R}}_{I}\left(t-\left(R_{-}+R_{+}\right)\right)}{1+{\dot{R}}_{I}\left(t-\left(R_{-}+R_{+}\right)\right)}\right)=F_{\mathrm{eff}}^{\prime }\left[t-\left(R_{-}+R_{+}\right)+R_{I}\left(t-\left(R_{-}+R_{+}\right)\right)\right],$$
which is satisfied because it is simply Equation (Equation 37) evaluated at $0<t^{\prime }=t-\left(R_{-}+R_{+}\right)<\tau$.

It is worth mentioning a couple of generalizations that can be derived from the previous result. The first one is that since the initial Moore function $F_{I}$ is $2R_{+}$-periodic, it can be extended to a linear function at $z_{n}=R_{-}+2nR_{+}$ for any natural *n*, which corresponds to applying the erasing trajectory $R_{\mathrm{II}}\left(t\right)$ at $t_{n}=R_{-}+\left(2n-1\right)R_{+}$. Therefore, so far, we have shown that any given initial trajectory $R_{I}\left(t\right)$ with a small enough duration ($\tau <R_{\pm }$) can be completed to an adiabatic shortcut by following the protocol
(47)$$R_{\mathrm{eff}}\left(t\right)=\left\{\begin{array}{cc} \begin{array}{c} R_{-}\\ R_{I}\left(t\right)\\ R_{+}\\ R_{\mathrm{II}}\left(t\right)\\ R_{-} \end{array} & \begin{array}{c} t<0\\ 0<t<\tau \\ \tau <t<t_{n}\\ t_{n}<t<t_{n}+\tau \\ t_{n}+\tau <t \end{array} \end{array}\right.,$$
with $R_{\mathrm{II}}\left(t\right)$ set by Equation (Equation 45).

A second generalization that can also illustrate the mechanism behind shortcut completion for short pulses is allowing the erasing trajectory to be executed by the left mirror. In this case, since initially the left mirror is at rest, we have L(0<t<τ)=0. Equations (Equation 5) and (6) then imply that $F_{I}=G_{I}$ and
(48)$$F_{I}\left(t+R_{I}\left(t\right)\right)-F_{I}\left(t-R_{I}\left(t\right)\right)=2\;.$$Therefore, the derivative of *F* satisfies Equation (Equation 37), and we can thus extend it smoothly in the same manner as before.

However, now the erasing trajectory has a static right mirror, $R_{\mathrm{II}}\left(t\right)=R_{+}$, and a nontrivial trajectory for the left mirror, $L_{\mathrm{II}}\left(t\right)$, to be determined by the Moore equations
(49)$$\begin{array}{cc} G_{\mathrm{eff}}\left(t+L_{\mathrm{II}}\left(t\right)\right)-F_{\mathrm{eff}}\left(t-L_{\mathrm{II}}\left(t\right)\right) & =0 \end{array}$$
(50)$$\begin{array}{cc} G_{\mathrm{eff}}\left(t+R_{-}\right)-F_{\mathrm{eff}}\left(t-R_{-}\right) & =2. \end{array}$$

The second equation implies $G_{\mathrm{eff}}\left(t\right)=2+F_{\mathrm{eff}}\left(t-2R_{-}\right)$ which, by replacing in the first one and taking the time derivative, leads to
(51)$$F_{\mathrm{eff}}^{\prime }\left(t+L_{\mathrm{II}}\left(t\right)-2R_{-}\right)=\frac{1}{R_{-}}\frac{\left[1-{\dot{L}}_{\mathrm{II}}\left(t\right)\right]}{\left[1+{\dot{L}}_{\mathrm{II}}\left(t\right)\right]}.$$This equation determines the erasing trajectory for the left mirror and can be solved by taking
(52)$$L_{\mathrm{II}}\left(t\right)=R\left(t-t_{n}\right)-R_{-},\;\;t_{n}<t<t_{n}+\tau$$
where $t_{n}=2nR_{+}+R_{-}$ for any natural number *n*. This can be seen simply by replacing it in the previous equation and comparing again with Equation (Equation 37).

In order to illustrate these results, we consider an initial trajectory given by Equation (Equation 34) with
(53)$$\begin{array}{cc} f(t) & =R_{-}\cos\left[A\sin^{2}\left(\omega t\right)\right], \end{array}$$
where *A* and ω are fixed parameters. As in the previous example, this choice ensures the continuity of $R_{I}\left(t\right)$ and its first derivatives. In Figure 5, we can see that by following the protocol given by Equation (Equation 47), i.e., executing the trajectory $R_{\mathrm{II}}$ at the precise time $t_{1}=2R_{-}$, the derivative Moore function remains constant, thereby erasing the photons generated by the pulse $R_{I}$ and producing an adiabatic shortcut.

To have a better understanding of how this is actually achieved, we can look at the energy density inside the cavity in Figure 6. There, we can observe that a pulse of energy is emitted, it then reflects off the left mirror and comes back to the right mirror precisely when the erasing trajectory $R_{\mathrm{II}}$ begins and reabsorbs it by moving in the opposite direction of the photons propagation. This mechanism can also be seen in the case of two moving mirrors. Once again, we see that the destructive interference necessary to erase the initially generated photons requires that during the second trajectory, the mirror should move in a direction opposite to the propagation of the pulse at the precise time when the pulse reaches that boundary (see Equation (Equation 52)). Therefore, one would expect that the adiabaticity parameter and thus the final state of the cavity would be highly dependent on the time $t_{n}$. This is indeed the case, as can be seen in Figure 6, where although we have Q=1 for $t=t_{1}=2R_{-},t_{2}=4R_{-}$, at intermediate times, *Q* deviates greatly from adiabaticity.

The advantage of this method over the previously presented is clear: for short pulses, one does not need to compute the Moore function; it is enough to apply the erasing trajectory for the right mirror $R_{\mathrm{II}}$ (or $L_{\mathrm{II}}$ for the left mirror) at any of the precise times $t_{n}$. This will annihilate the particles and execute an STA that will return the state of the cavity to the ground state with the same initial length.

### 4.3. Time Inversion

In this section, we present a third method for completing STA by taking advantage of a time-inversion symmetry of the system. As a byproduct of this approach, we show that the time reversal of an STA is also an STA, which is a result that may be useful when considering thermodynamic cycles.

First, we note that although the physical theory does not have time-inversion invariance due to the moving boundary condition, the theory in the conformal variables Equation (Equation 7) is symmetric with respect to the time inversion $\bar{t}\to -\bar{t}$. It is simple to see that this symmetry in the conformal variables generates a symmetry in the physical theory given by the transformation t→−t, i.e., a time reversal, and F(z)→−G(−z),G(z)→−F(−z). We call this conformal time-reversal symmetry.

From this, we can establish two results. One is that even when a trajectory L(t),R(t) is not an STA for the field, if there are constants $\tau ,\tilde{z},C_{F},C_{G}$ such that the Moore functions satisfy
(54)$$\begin{array}{c} F\left(z\right)=-G\left(-z+\tilde{z}\right)+C_{F}\;z>>\tau \end{array}$$
(55)$$\begin{array}{c} G\left(z\right)=-F\left(-z+\tilde{z}\right)+C_{G}\;z>>\tau , \end{array}$$
then there are times at which implementing L(t),R(t) followed by its temporal reverse L(−t),R(−t), generates an STA. This is because, under these conditions, it is possible to smoothly continue the Moore functions of the trajectories with those associated with their temporal reverse. Since the former are initially linear, the latter must also be linear at late times. Therefore, a smooth continuation of the first Moore function into the other generates an STA formed by the trajectory followed by its temporal reversal.

In the case that L(t)=0, we have F(z)=G(z), and the previous condition can be expressed more simply as
(56)$$F^{\prime }\left(z\right)=F^{\prime }\left(-z+\tilde{z}\right)\;z>>\tau ,$$
meaning that the derivative of the Moore function at late times should be an even function for some suitable choice of the origin. In fact, since the Moore function is $R_{+}$-periodic if $\tilde{z}$ satisfies this condition, then ${\tilde{z}}_{n}=\tilde{z}+2nR_{+}$ also does, and so to complete a shortcut the time-reversed trajectory has to be implemented at certain discrete times $t_{n}$.

This method works only when the complete shortcut starts and ends at the same position. However, it works for any initial evolution, i.e., it does not need to be a short pulse. It can have any duration as long as the Moore function verifies the condition given by Equation (Equation 54).

In order to check this result numerically, we consider the initial trajectories $R_{I}$ given by Equations (Equation 34) and (Equation 35), and then we apply it to the cavity followed by the time-reversed trajectory at times $t_{n}$, carefully chosen so that the two Moore functions coincide to form a longer trajectory $R_{\mathrm{eff}}$. As we can see in Figure 7, acting on the time-reverse trajectory at discrete times, we manage to take the adiabaticity parameter *Q* from 0.6 back to 1, signaling a successful adiabatic shortcut.

A second result that can be obtained from the conformal time-reversal symmetry is that if the trajectories L(t),R(t) constitute an STA, then the time-reversed trajectories are also an STA. To see that this is the case, note that the Moore functions of the original STA are linear both at early and late times
(57)$$F\left(z\to \pm \infty \right)=\frac{z}{R_{\pm }-L_{\pm }}+\frac{1}{2}\frac{R_{\pm }+L_{\pm }}{R_{\pm }-L_{\pm }}$$
(58)$$G\left(z\to \pm \infty \right)=\frac{z}{R_{\pm }-L_{\pm }}-\frac{1}{2}\frac{R_{\pm }+L_{\pm }}{R_{\pm }-L_{\pm }}.$$Therefore, applying a conformal time-reversal transformation, we can conclude that the time-reversed trajectories L(−t),R(−t) have reverse Moore functions given by
(59)$$F_{\mathrm{rev}}\left(z\to \pm \infty \right)=\frac{z}{R_{\mp }-L_{\mp }}+\frac{1}{2}\frac{R_{\mp }+L_{\mp }}{R_{\mp }-L_{\mp }}$$
(60)$$G_{\mathrm{rev}}\left(z\to \pm \infty \right)=\frac{z}{R_{\mp }-L_{\mp }}-\frac{1}{2}\frac{R_{\mp }+L_{\mp }}{R_{\mp }-L_{\mp }},$$
and so these trajectories are also adiabatic shortcuts but with initial and final positions exchanged. This property is particularly useful when considering thermodynamic cycles, in which we need not only an STA for the position of the boundaries to change from $L_{-},R_{-}$ to $L_{+},R_{+}$ but also during a second stroke that returns the mirrors to their original positions.

We can illustrate this numerically considering a reference trajectory given by Equations (Equation 34) and (Equation 35) and use them to calculate the corresponding effective STA following Section 2.1, $R_{\mathrm{eff}}\left(t\right)$, which starts at $t_{i}$ and ends at $t_{f}$. We then compute the temporal reverse $R_{\mathrm{rev}}\left(t\right)=R_{\mathrm{eff}}\left(t_{f}-t\right)$ and the adiabaticity coefficient $Q_{\mathrm{rev}}$. We can clearly see from Figure 8 that $Q_{\mathrm{rev}}$ starts equal to 1, oscillates, and goes back to unity when the motion stops signaling that the evolution was indeed an adiabatic shortcut.

As already mentioned, this property is also valid for a harmonic oscillator with time-dependent frequency: the temporal reverse of an STA is also an STA.

## 5. Discussion

The fast manipulation of micromachines can result in excessive losses in the form of quantum friction, which reduces its efficiency. Thus, finding an STA is of paramount importance in quantum thermodynamics since, even after considering the energy cost associated, STA-enhanced thermal machines have the potential for greatly enhanced efficiency.. It also allows for the design of rapid quantum gates, which makes it useful for quantum information processing as well. Here, we propose an additional application of an STA for state preparation. Once an initial state has been prepared, if a perturbation of the parameters of the system change suddenly, it will greatly reduce the fidelity of the desired state; however, knowing how they have been changed, we can complete this time variation into an STA which, by definition, will restore the state of the system to the initial one. This method can then allow for longer times between the preparation of a state and its processing [42].

In this paper, we addressed the problem of how to transform an initially nonadiabatic evolution into an STA, for a quantum field confined within a one-dimensional cavity featuring two moving mirrors. As outlined in our previous studies, the state of the field is determined by two Moore functions, *F* and *G*. In a time interval where the cavity remains static, these functions can be expressed as the sum of a linear and a periodic function. The presence of a nonzero periodic component indicates an excited state of the field. By characterizing the field’s state using *F* and *G*, we can employ reverse engineering techniques to construct an STA. Regardless of the initial evolution, it is possible to smoothly extend the Moore functions to asymptotically linear functions, from which we can compute the mirrors trajectories.

We identified an inspiring similarity between the Moore functions that describe the state of the field in the one-dimensional cavity and the Ermakov function ρ that provides a formal solution for the quantum harmonic oscillator with a time-dependent frequency. Within a time interval of constant frequency, $\rho^{2}$ can be expressed as the sum of a constant and a periodic function. When the periodic component is nonzero, it describes an oscillator in a squeezed state. The STA completion can be constructed by extending the Ermakov function in such a manner that ρ tends to a constant as time approaches infinity.

We presented three different methods for completing an STA for the optomechanical cavity. All of them are based on the fact that the Moore functions should be linear before and after the perturbation in order for a trajectory of the mirrors to comprise an adiabatic shortcut.

The first method, presented in Section 4.1, consists of three steps: (1) computing the Moore functions, (2) extending them smoothly into a linear functions, and then (3) using them to compute the trajectory using inverse engineering. This method is conceptually simple, it can be used for any perturbation and executes an STA for any final position of the mirror; however, the three steps require implementing precise measurements or demand challenging computations.

The second method, described in Section 4.2, addresses these problems and solves them in the case that the perturbation consists of a short pulse. In that case, the second trajectory, which erases the excitations and restores the state of the system, can be given explicitly in terms of the initial perturbation. This allows us to avoid the computation and inversion of the Moore function entirely. The only restrictions are that perturbation time must be short and the final length of the cavity should be the same as the initial one.

The third method, presented in Section 4.3, takes advantage of a symmetry of the system to show that the time-reversed trajectory of the perturbation can be used to complete an adiabatic shortcut if the derivative of the Moore function of the perturbation is even for late times. In this case, the perturbation can have an arbitrary duration, but it is not necessary to compute its Moore function to check the parity. Nonetheless, when comparing this method with the first one, we avoid extending it smoothly and inverting it to compute the trajectory, which constitutes a great simplification.

With regard to the possible experimental implementation, it should be noted that the dynamical Casimir effect has been effectively demonstrated in superconducting circuits more than a decade ago [43]. In that case, a SQUID was used instead of a mechanical moving mirror, which simulated a time-dependent boundary condition for the field. The protocols presented here could be implemented in this type of setup by making use of two SQUIDs at both ends of a superconducting cavity in a way similar to [44]. Additionally, it was recently proposed to measure the dynamical Casimir effect due to true mechanical motion by using film bulk acoustic resonators in the GHz spectral range [45], which could also lead to an implementation of the schemes proposed here.

Further investigations should be conducted in order to establish how the different methods discussed here to complete an STA can be adapted for other systems, as well as how much they could improve the time delay between state preparation and processing under experimental conditions.

## Figures and Tables

**Figure 1 entropy-25-01249-f001:**
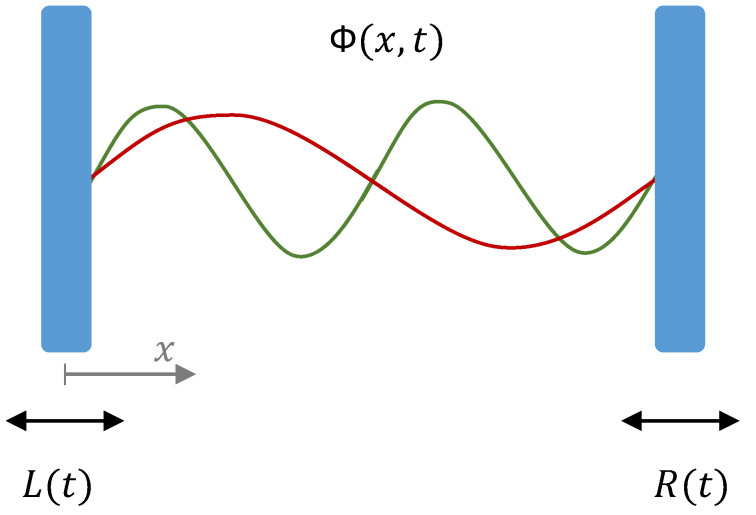
Schematics of the one-dimensional cavity with a scalar quantum field Φ(x,t) inside and two moving mirrors with trajectories L(t) and R(t): The red and green curves illustrate two of the infinite modes of the field in the cavity.

**Figure 2 entropy-25-01249-f002:**
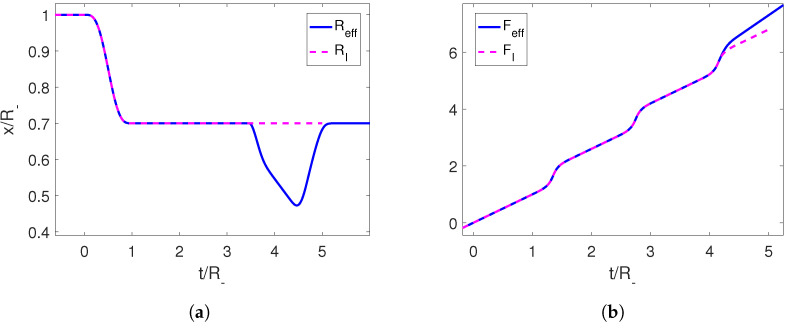
(**a**) Initial trajectory for the right mirror in magenta and its completion to an adiabatic shortcut in blue. (**b**) Moore functions for the corresponding trajectories. The parameters employed for the first trajectory are $R_{-}=1$, ϵ=0.3 and τ=1.

**Figure 3 entropy-25-01249-f003:**
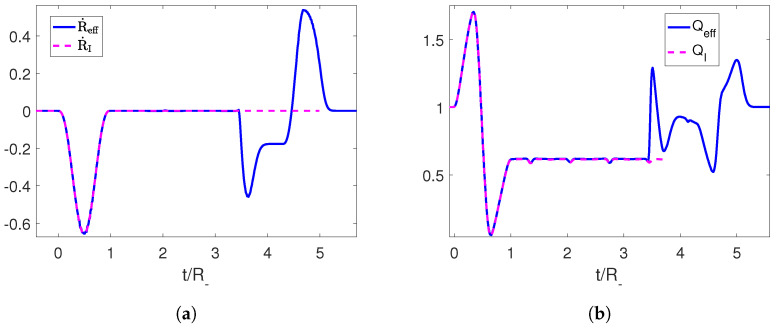
(**a**) Velocity for the initial trajectory of the mirror and for its completion to an adiabatic shortcut in blue. (**b**) Adiabaticity parameters for the corresponding trajectories. The parameters employed for the first trajectory are $R_{-}=1$, ϵ=0.3 and τ=1.

**Figure 4 entropy-25-01249-f004:**
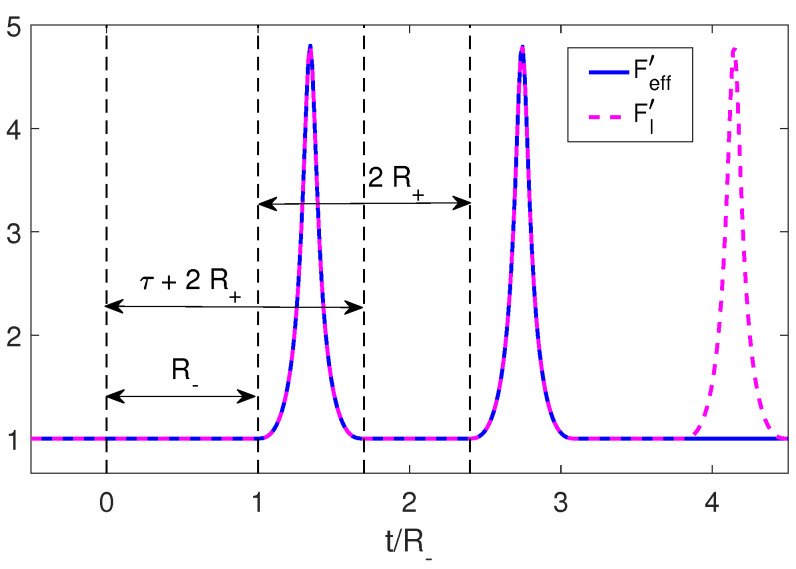
Structure of the derivative of the Moore function for a brief pulse trajectory in dashed magenta and the completion to an adiabatic shortcut in solid blue line.

**Figure 5 entropy-25-01249-f005:**
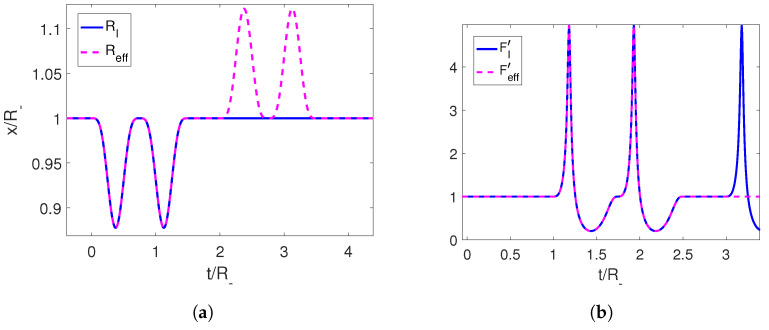
(**a**) Trajectory for the initial velocity of the mirror in blue and for its completion to an adiabatic shortcut in magenta. (**b**) Derivative of the Moore function for the corresponding trajectories. The parameters employed for the first trajectory are $R_{-}=R_{+}=1$, A=0.5 and ω=2π/τ=2π/1.5.

**Figure 6 entropy-25-01249-f006:**
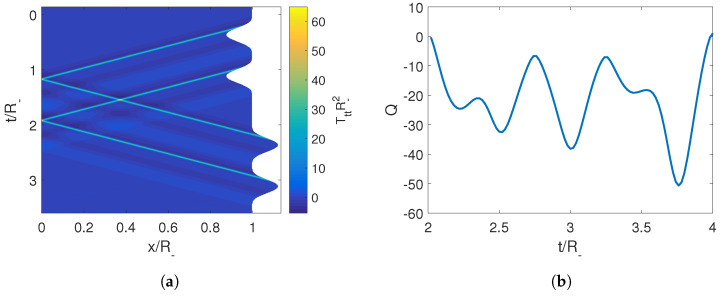
(**a**) Energy density of the field inside the cavity for the effective trajectory composed of an initial trajectory followed by an erasing trajectory at $t_{1}$. (**b**) Adiabaticity parameter obtained when by implementing the erasing trajectory Equation (Equation 45) at different times $t_{n}$. Note that Q=1 for $t=t_{1}$ and $t=t_{2}$. The parameters employed for the first trajectory are $R_{-}=R_{+}=1$, A=0.5, ω=2π/1.5, $t_{1}=2R_{-}$.

**Figure 7 entropy-25-01249-f007:**
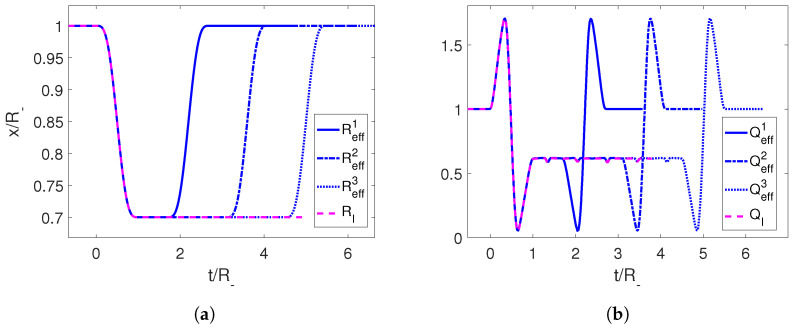
(**a**) First trajectory for the mirror in magenta and the effective trajectories $R_{\mathrm{eff}}^{n}$ using its temporal reverse at different times $t_{n}$ in blue. (**b**) Adiabaticity parameters for the corresponding trajectories. The parameters employed for the first trajectory are $R_{-}=1$, ϵ=0.3 and τ=1.

**Figure 8 entropy-25-01249-f008:**
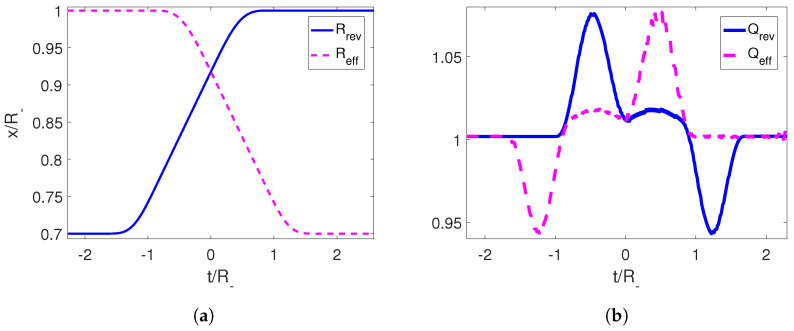
(**a**) Trajectories for the right mirror for an adiabatic shortcut in magenta and its temporal reverse in blue. (**b**) Adiabaticity parameters for the adiabatic shortcut and its temporal reverse. The parameters employed for the effective shortcut trajectory aew $R_{-}=1$, ϵ=0.3 and τ=1.

## Data Availability

Not applicable.

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
