# Peer review of "Adiabatic Shortcuts Completion in Quantum Field Theory: Annihilation of Created Particles"

_entropy, 2023, doi:10.3390/e25091249_

Round 1

Reviewer 1 Report

The authors consider an interesting topic, related with the notion of shortcuts to adiabaticity (STA) in quantum field theory, in particular in the relevant system of a cavity with moving boundaries. The idea is to transform an initially non-adiabatic evolution into an adiabatic one -the authors refer to this process as completing an STA. In a cavity with moving boundaries all this have a very nice connection with the Dynamical Casimir effect, which is a highly non-adiabatic process. Therefore, completing the STA turns out to be equivalent to “erase” the initially emitted particles. 

The article is very well written, including pedagogical descriptions and examples. The methods and analysis are sound. Results are potentially useful in several branches of the interdisciplinary community of modern quantum technologies. While it might have been desirable some discussion of the feasibility of the proposed effective mirror trajectories in a realistic experimental setup, I respect the authors approach of keeping the results mostly in an abstract fashion. 

I am happy to recommend this paper for publication in the Special Issue of Entropy in its current form.

Author Response

We thank the referee for his positive view of our manuscript. We have added a paragraph at the end of the Discussion Section about the feasibility of the proposed effective mirror trajectories in a realistic experimental setup.

Sincerely,

F. Lombardo

Reviewer 2 Report

In this paper, the authors investigated the completion of a non-adiabatic evolution into a shortcut to adiabaticity for a quantum field confined within a one-dimensional cavity containing two movable mirrors. I think it is interesting to find a STA for the quantum field in a moving quantum cavity. The method is clearly presented. The results look nice. I only have two main questions:

1) It seems that the STA trajectories of the mirrors work well in the system in a finite time, I wonder what's the energy cost of quantum system during the non-adiabatic evolution? Is it possible to see the quantum speed limit in this system?

2) Is it possible to have an experimental proposal of the shortcut which I would suggest to add it for the discussion section?

and some minors:

1) Some references contain titles but some do not. Please refer to the journal latex template.

2) In the introduction section, I think that single and many-particle quantum heat engine have more references, for instance,

T. Keller, T. Fogarty, J. Li, T. Busch, Phys. Rev. Research 2, 033335(2020);

J. Li, T. Fogarty, S. Campbell, X. Chen, T. Busch, New J Phys. 20, 015005 (2018);

Minor editing of English language required. 

Author Response

We thank the referee for her/his positive view of our manuscript. Regarding the questions asked by the referee:

1) Although these protocols can be used to dramatically improve the efficiency of a thermodynamic cycle, the energy cost of the shortcut should be taken into account. While there is no universal consensus on exactly how to measure this cost, there are possible measurements in standard quantum mechanics. However, in quantum field theory previous measures should be adapted. Further investigations are needed in order to understand whether this is in fact a faithful measures. We will work on that soon. We have added a paragraph with the necessary equations for the measurement of cost in quantum mechanics, in section 2.1.

2) We added a discussion in the last Section. 

We also corrected references and added the ones suggested by the referee.

Sincerely,

F. Lombardo

Reviewer 3 Report

The quality of the English is fine, with only a few minor typos/grammatical errors.

Author Response

We thank the referee for his very positive view of our manuscript.  We have revised the paper by giving a better presentation, and addressing specific questions raised by the referees. Below we present a detailed answer to all the criticisms contained in the reports.

  • We have added a references to different methods for obtaining shortcuts to adiabaticity (such as transitionless quantum driving, optimal protocols, etc) in the Introduction.
  • We have added the suggested references for single and many-particle quantum heat engines.
  • We have added a dicussion on the cost of adiabatic shortcuts in quantum field theory in Section 2.1.
  • We have added an experimental proposal of the shortcut to the discussion section.
  • We have added a suggested reference on the aplication of STAs to speed up quantum gates.

We hope that you find our changes satisfactory, and the manuscript suitable for 
publication.

Sincerely,

F. Lombardo

Round 2

Reviewer 2 Report

The manuscript has been improved a lot. I would like to recommend the manuscript to publish in journal Entropy.

Author Response

Thanks to the referee for his/her consideration

Reviewer 3 Report

In the revised manuscript the authors have satisfactorily my comments and I am happy to recommend publication in Entropy. I only have one additional suggestion. The new text added to the first paragraph of the discussion states:

"Thus finding STA is of paramount importance in quantum thermodynamics since, even after considering the energy cost associated, it greatly improves the efficiency."

Since, as the authors note below Eq. 17 of the revised manuscript, a full accounting of the energetic cost has not been established, I recommend modifying this statement to say something along the lines of, "...even after considering the energy cost associated, STA-enhanced thermal machines have the potential for greatly enhanced efficiency."

Author Response

We thanks the referee for the comment. We have followed his/her recommendation and changed the sentence as he/she suggested.

Sincerely,

F. Lombardo